# OpenReview forum: "Breaking the Exploration Bottleneck: Rubric-Scaffolded Reinforcement Learning for Open-Ended LLM Reasoning"
_ICML.cc/2026/Conference — ICML 2026 regular_

### Official Review · Reviewer_ndzB · 2026-02-26

**Soundness:** 3
**Presentation:** 3
**Significance:** 3
**Originality:** 3
**Overall Recommendation:** 3
**Confidence:** 4

**Summary:**

This paper proposes Rubric-Scaffolded Reinforcement Learning (RuscaRL) to address an “exploration bottleneck” in RLVR for open-ended reasoning tasks. The key idea is to use checklist-style rubrics in two roles: (i) explicit scaffolding during rollout generation by injecting a subset of rubric criteria into the instruction prompt, with intra-group scaffolding differentiation and inter-step scaffolding decay; and (ii) verifiable rewards by having a grader LLM do criterion-wise binary judgments and aggregating criterion points into a scalar reward normalized by the total positive points. Experimentally, the authors claim consistent gains across medical, writing, and instruction-following benchmarks, and present analyses arguing that RuscaRL increases diversity early and discovers high-“novelty” trajectories compared to a rubric-reward RL baseline.

**Compliance With Llm Reviewing Policy:**

Affirmed.

**Final Justification:**

See reply to rebuttal.

**Key Questions For Authors:**

See above.

**Strengths And Weaknesses:**

S1. Rubric-based reward formulation is simple and scalable. The grader performs binary checks per criterion, and the reward is a normalized sum (with negatives possible), which is a pragmatic approach for open-ended tasks lacking ground-truth verification.

S2. Thoughtful empirical probing of exploration behavior. The paper goes beyond aggregate benchmark scores by reporting diversity evolution (Self-BLEU / semantic distance plots) and a “novelty” analysis based on extreme importance ratios and associated score improvements.

S3. Robustness analysis to rubric corruption/noise is a good direction. The construction of “Inverse / Negated / Ambiguous / Contradictory / 50% removed” rubric variants and side-by-side training is valuable and not commonly included.

W1. You set (\lambda_i=\frac{G-i}{G-1}), so the last sample in each group has (\lambda=0) regardless of training stage, and early training stages also include heavily scaffolded samples. This means each GRPO group mixes trajectories from different behavior policies: (\pi(\cdot\mid q,R_{S,i})) for different (i). Yet advantages are normalized within the same group. Do you observe that the advantage distribution is skewed by (\lambda) (e.g., (A_i) monotonically increases with (\lambda_i))? If so, how do you prevent updates from over-emphasizing scaffolded trajectories?

W2. For datasets without ideal answers, you first generate example solutions with GPT-4.1 and then generate rubrics based on those solutions using the template in Box E.3. This can inadvertently align rubrics to the style/structure of GPT-4.1 answers (and the rubric generator prompt’s priors), making it easier for RuscaRL to learn “what the rubric-writer likes” than to learn task-grounded competence. Can you provide at least one evaluation where rubrics are authored by a different generator model/prompt, or where the judge is replaced by a rubric set independent of the GPT-4.1-generated exemplars?

W3.  Appendix D.6 frames the theoretically correct per-token ratio for training a no-scaffold target policy. However, Table 9 shows that the paper’s preferred method (matching numerator/denominator without (R_S)) performs best while being “not a true importance sampling correction.” If the empirically best ratio is not unbiased, what objective is RuscaRL actually optimizing? Can the authors formalize it as a KL-regularized objective toward a no-scaffold reference (as hinted in D.6)?


W4. The title is about 'General LLM Reasoning' but the authors didn't eval on any popular reasoning tasks. I suggest slightly revising the claim.

---

> ### Author Rebuttal · Authors · 2026-03-31
>
> We thank the reviewer for recognizing several positive aspects of our work, including the simplicity and scalability of the rubric-based reward design, the analysis of exploration behavior, and the robustness study under noisy or corrupted rubrics.
> Below we address the main points raised in the review.
>
> **[W1] Over-emphasizing scaffolded trajectories**
>
> Thanks for this insightful question.
> We do observe a positive correlation between scaffolding strength and reward/advantage, as shown in Table R1.
> We view this as an intended property rather than a pathology, since stronger scaffolding is designed to provide higher-quality proposal trajectories in early training.
> More importantly, the scaffolding ratio decays over training, so the optimization gradually shifts from guided exploration to the no-scaffold setting.
> Our ablations also show that keeping all samples fully scaffolded performs worse than our differentiated-and-decayed design, indicating that persistent reliance on scaffolded trajectories is not what drives the gains.
> Consistent with this, we observe sustained improvement on unscaffolded inference in late training, suggesting that the model internalizes the reasoning patterns instead of merely following prompt hints.
>
> Table R1. Reward and Advantage Distribution across Scaffolding Levels
> |Sample Index ($i$)|Score (Mean±Std)|Advantage (Mean±Std)|Positive Advantage Ratio|
> |:--:|:--:|:--:|:--:|
> |0|0.564±0.324|0.238±0.881|64.06%|
> |1|0.572±0.290|0.202±0.853|59.38%|
> |2|0.569±0.320|0.237±0.931|62.50%|
> |3|0.546±0.311|0.171±0.967|54.69%|
> |4|0.498±0.336|-0.111±1.073|43.75%|
> |5|0.482±0.295|-0.076±0.861|46.88%|
> |6|0.476±0.331|-0.196±0.938|39.06%|
> |7|0.434±0.304|-0.466±1.102|31.25%|
>
> **[W2] Rubric alignment to GPT-4.1 style / Generator prompt priors**
>
> Thanks for raising this concern.
> Our evaluation already uses independent judges: Qwen3-32B is used only for training-time grading; following their respective original settings, HealthBench and LLMEval-Med are evaluated by GPT-4.1, while WritingBench and Creative Writing are evaluated by Claude-Sonnet-4.
> To further test whether RuscaRL learns task-grounded competence rather than a specific rubric prior, we additionally train Qwen2.5-7B-Instruct on RubricHub, an open-source dataset whose rubrics are generated by aggregating multiple models, including Gemini 3 and GPT-5.1.
> As shown in Table R2, RuscaRL still clearly outperforms Rubric-based RL.
> This suggests that the gains are not tied to a single rubric generator or prompt prior, and that our scaffolding framework transfers across heterogeneous rubric sources.
>
> Table R2. Performance of Qwen2.5-7B-Instruct trained on the open-source RubricHub dataset.
> |Model|HealthBench|LLMEval-Med|
> |:--|:--|:--|
> |Qwen2.5-7B-Instruct (Initial)|23.4|48.0|
> |+ Rubric-based RL|50.1|54.4|
> |+ RuscaRL (Ours)|55.9|64.8|
>
> **[W3] Importance sampling method analysis**
>
> Thank you for the insightful question.
> We agree our ratio $\\rho\_t^{\\text{NS}}(\\theta)=\\frac{\\pi\_\\theta(o\_t \\mid q, o\_{ < t })}{\\pi\_{\\theta\_{\\text{old}}}(o\_t \\mid q, o\_{ < t })}$ is not an unbiased estimator of the exact no-scaffold objective. The theoretically unbiased correction is $\\tilde{\\rho}\_t^{\\text{NS}}(\\theta)=\\frac{\\pi\_\\theta(o\_t \\mid q, o\_{ < t })}{\\pi\_{\\theta\_{\\text{old}}}(o\_t \\mid q, \\mathcal{R}\_S, o\_{ < t })}$.
>
> Instead, RuscaRL optimizes a PPO/GRPO-style proximal surrogate:
>
> $$\\mathcal{L}\_{\\text{RuscaRL}}(\\theta)=\\mathbb{E}\_{q, o \\sim \\pi\_{\\theta\_{\\text{old}}}(\\cdot \\mid q, \\mathcal{R}\_S)}\\left[\\sum\_t \\min\\Big(\\rho\_t^{\\text{NS}}(\\theta)\\hat{A}(q,o), \\text{clip}(\\rho\_t^{\\text{NS}}(\\theta), 1-\\epsilon, 1+\\epsilon)\\hat{A}(q,o)\\Big)\\right]$$
>
> This maximizes rubric rewards on scaffolded rollouts while regularizing toward the previous no-scaffold policy, acting as a KL-regularized objective rather than an exact IS correction.
>
> We prefer this surrogate for stability. The theoretically unbiased ratio uses mismatched conditioning contexts, leading to high-variance IS weights and unstable training. Table 9 reflects this bias-variance trade-off: our matched-context variant trades theoretical unbiasedness for practical stability, effectively encouraging the policy to internalize scaffold-free reasoning.
>
> **[W4] Title over-claiming ("General LLM Reasoning")**
>
> Thanks for the constructive suggestion.
> We agree that the main empirical focus of this paper is indeed on open-ended reasoning tasks, where our method is most directly motivated and most extensively evaluated.
> To better reflect this focus and avoid over-claiming, we will revise the title to emphasize "Open-Ended LLM Reasoning" in the camera-ready version.
> For completeness, we would also like to briefly note that RuscaRL does show consistent improvements on traditional reasoning tasks (including MATH, AMC, and AIME) as well, which we included in Appendix D.2 (Figure 5).

---

> > ### Author Rebuttal · Reviewer_ndzB · 2026-04-02
> >
> > Thank you for the detailed and thoughtful rebuttal. I appreciate the additional analysis provided for W1, particularly the quantitative breakdown of advantage distribution across scaffolding levels and the clarification that decay mitigates over-reliance on scaffolded trajectories. The additional experiments in W2 using heterogeneous rubric sources (e.g., RubricHub) also help alleviate concerns about overfitting to a single rubric generator.
> >
> > For W3, the clarification that the method optimizes a PPO/GRPO-style KL-regularized surrogate rather than an unbiased importance sampling objective is helpful, and I agree this framing better explains the empirical stability.
> >
> > While these clarifications strengthen the work, some concerns remain regarding the interaction between scaffolding heterogeneity and group-relative optimization. Therefore, I will maintain my score.

---

> > > ### Author Response · Authors · 2026-04-06
> > >
> > > Thank you for the detailed follow-up. We appreciate that our rebuttal helped clarify the main concerns and strengthen the work overall. We would like to provide a further clarification on the interaction between scaffolding heterogeneity and group-relative optimization.
> > >
> > > Our empirical evidence suggests that the gain does not come from over-emphasizing heavily scaffolded trajectories. If that were the case, then the most heavily scaffolded settings should perform best. However, our ablations show the opposite.
> > >
> > > In Figure 4a, the differentiated Linear design achieves the best result (56.4), outperforming both Binary (N=0) (52.0) and the fully scaffolded Binary (N=8) (44.7). Similarly, in Figure 4b, Sigmoid decay performs best (56.4), while Constant full scaffolding is much worse (44.0). This shows that persistent reliance on scaffolded trajectories is not what drives the improvement.
> > >
> > > We therefore believe the evidence supports the following interpretation: RuscaRL benefits from differentiated and decayed scaffolding, which improves exploration early on while avoiding over-reliance on prompts later in training. We will clarify this point more explicitly in the final version.

---

### Official Review · Reviewer_cuDF · 2026-03-13

**Soundness:** 2
**Presentation:** 3
**Significance:** 2
**Originality:** 3
**Overall Recommendation:** 3
**Confidence:** 4

**Summary:**

This paper proposes RuscaRL, a reinforcement learning framework inspired by instructional scaffolding theory, designed to break the exploration bottleneck in RLVR training for LLMs. This paper addresses a central concept of using checklist-style rubrics in two complementary ways — as explicit scaffolding during rollout generation and as verifiable rewards during training (via LLM-as-a-Judge binary evaluation) — thereby extending RLVR to open-ended tasks such as medical consultation, creative writing, and instruction following. Experiments across multiple model families and benchmarks demonstrate that RuscaRL consistently outperforms strong baselines, with a Qwen3-30B-A3B-Instruct model matching OpenAI-o3 performance on HealthBench-500.

**Compliance With Llm Reviewing Policy:**

Affirmed.

**Final Justification:**

While the rebuttal clarifications strengthen the work, certain concerns remain about using the same rubrics across rollout generation and evaluation criteria for answer leakage problem and the limited testing to other tasks such as deep research tasks, long-context reasoning tasks and general reasoning tasks such as code and agent planning, tool use abilities.

**Key Questions For Authors:**

1. Have you considered dynamically updating or expanding rubrics as the model improves during training? How would static rubrics perform when the model has already mastered most criteria?
2. Since the same rubrics are used for both rollout guidance and reward evaluation, how do you ensure the model is learning genuine reasoning rather than superficially gaming the rubric criteria? Can you provide evidence distinguishing the two?
3. Grader Bias: Given that the same grader model (Qwen3-32B) is used for both training rewards and final evaluation, have you measured human-grader agreement or tested with an independent grader to rule out systematic bias in favor of RuscaRL?
4. The optimal sigmoid parameters (α, t₀) vary across model families. What is the recommended strategy for selecting these hyperparameters when applying RuscaRL to a new model, and how sensitive is performance to suboptimal choices?

**Strengths And Weaknesses:**

Strength
1. The authors propose using rubrics not only as reward signals but also as explicit instructional scaffolding during rollout generation. The inspiration from Vygotsky's scaffolding theory is well-articulated, and the combination of intra-group differentiation and inter-step sigmoid decay provides an elegant unified mechanism to promote diversity while preventing overfitting to external guidance.
2. The paper is clearly written and well-structured.
3. The empirical evaluation is extensive, covering multiple model families, varying parameter scales, and diverse task domains (medical, writing, instruction following, STEM), supplemented by thorough ablation studies, Best-of-N analysis, novelty/diversity metrics, and robustness tests, providing a well-rounded assessment of the proposed method.

Weakness
1. Since the same rubrics are used both as scaffolding during rollout generation and as evaluation criteria for reward computation, there is a potential "answer leakage" concern — the model is essentially told what will be evaluated before generating its response. This may lead the model to superficially satisfy rubric criteria rather than developing genuine reasoning capabilities.
2. The rubrics used in RuscaRL are fixed throughout training. As the model's capability improves over time, static rubrics may become insufficiently challenging, potentially creating a ceiling effect. A co-evolutionary rubric mechanism that progressively refines or expands criteria alongside model improvement could be more effective, but this direction is neither explored nor discussed.
3. The entire framework depends on the availability of well-designed rubrics. The paper briefly acknowledges this in the conclusion but does not provide a systematic analysis of how rubric quality affects performance, leaving a critical practical bottleneck unaddressed.
4. Limited Gains Outside the Primary Domain. Performance improvements on non-medical benchmarks are notably smaller, and in some cases negative (e.g., MedQA, MedMCQA for base models), suggesting that the method's effectiveness may be largely driven by the HealthBench training data rather than a general exploration improvement.
5. The framework critically depends on the LLM-based grader for both reward computation and evaluation, yet its reliability is not systematically validated. Key aspects such as human agreement, inter-rater consistency, and the impact of grader model choice on training stability are not analyzed.

---

> ### Author Rebuttal · Authors · 2026-03-31
>
> We sincerely thank the reviewer for the positive assessment of our work, particularly the originality of our rubric-based approach, the clarity of the presentation, and the broad empirical evaluation.
> Below we address the main points raised in the review.
>
> **[W1,Q2]: Shallow Compliance Risk**
>
> Thanks for this important concern. We mitigate the risk of gaming the rubric criteria in two ways.
>
> (1) Rubric scaffolding in RuscaRL is temporary and gradually decays during training, encouraging the model to internalize useful reasoning patterns rather than rely on explicit checklist hints.
>
> (2) Rewards are based on multi-dimensional rubric evaluation rather than simple rule matching, so superficial shortcuts that satisfy one criterion are penalized by failures on others. This is also supported by our case studies: Appendix G.1 shows that simple rule-based verification can reward keyword spamming, while Appendix G.2 provides a manual qualitative analysis comparing the initial and RuscaRL-trained responses, showing that RuscaRL improves the actual quality of the answer, rather than merely gaming the rubric criteria.
>
> **[W2,Q1]: Dynamic Update of Rubrics**
>
> We thank the reviewer for this insightful suggestion.
> We view dynamic rubric updating (e.g., OnlineRubrics, DR TULU and RLCER) and RuscaRL as orthogonal: the former improves rubric quality, while RuscaRL improves utilization.
>
> While a promising direction, our preliminary DeepResearch experiments (Table R1) showed that expanding criteria online can force hallucinations to satisfy new constraints, reducing factuality.
> To avoid these uncontrollable variables and the ceiling effect, we currently focus on pre-generating highly discriminative, challenging rubrics.
>
> Table R1. Comparison of baseline and dynamic updating rubrics on a DeepResearch task.
> |Method|Point Coverage|Factuality|
> |:--|:--|:--|
> |Baseline|54.7|66.1|
> |Dynamic Update Rubric|57.3|64.4|
>
> **[W3]: Impact of Rubric Quality on Performance**
>
> Thanks for the suggestion.
> We have extensively analyzed the impact of rubric quality on performance in Appendix D.8 of the paper. As shown in Table 11, RuscaRL maintains robustness against mild perturbations, though severe corruptions naturally degrade performance.
>
> **[W4]: Concerns on Gains Benefiting from HealthBench**
>
> Sorry for the confusion.
> RuscaRL's gains are not limited to medical tasks or the HealthBench dataset.
> * **Non-Medical Gains** Table 1 shows significant performance improvements in both the Writing and Instruction Following domains.
> * **MedQA/MedMCQA Results:** As noted in Section 5.2, these are closed-ended multiple-choice benchmarks, whereas our training targets open-ended tasks. We included them solely to demonstrate cross-task generalization, making marginal gains expected.
> * **Dataset Independence:** To verify that our scaffolding framework, not just the HealthBench data, drives the exploration improvements, we trained Qwen2.5-7B-Instruct using the open-source RubricHub dataset instead. As shown in Table R2, RuscaRL still substantially outperforms the Rubric-based RL baseline.
>
> Table R2. Performance when training with the RubricHub dataset.
> |Model|HealthBench|LLMEval-Med|
> |:--|:--|:--|
> |Qwen2.5-7B-Instruct|23.4|48.0|
> |+ Rubric-based RL|50.1|54.4|
> |+ RuscaRL|55.9|64.8|
>
> **[W5,Q3]: Grader Consistency and Training Stability**
>
> Sorry for the confusion.
> We would like to clarify that Qwen3-32B is not used across all pipelines. It is used as the grader during training, whereas the final evaluations use GPT-4.1 for HealthBench and LLMEval-Med, and Claude-Sonnet-4 for WritingBench and Creative Writing, as detailed in Appendix C.2.
> We have also added experiments assessing the impact of different grader models when training Qwen2.5-7B-Instruct, together with their agreement with human expert judgments on 940 expert-annotated criteria (Table R3).
> As expected, stronger graders generally improve both human agreement and downstream performance, but the gains plateau at larger scales.
>
> Table R3. Impact of grader model choice on medical performance and human agreement.
> |Grader|HealthBench|LLMEval-Med|Cohen's Kappa|F1 Score|
> |:--|:--|:--|:--|:--|
> |Qwen2.5-7B-Instruct|45.3|56.6|0.58|0.81|
> |Qwen3-30B-A3B-Instruct|48.9|60.9|0.74|0.90|
> |Qwen3-32B (non-thinking)|56.4|65.3|0.76|0.90|
> |gpt-oss-120B (auto)|56.6|65.0|0.74|0.88|
> |Qwen3-235B-A22B-Instruct|56.6|65.8|0.80|0.91|
>
> **[Q4]: Parameter Tuning Strategy and Sensitivity Analysis**
>
> Thanks for the valuable comment.
>
> * **Strategy**: Monitor policy entropy. Initially, RuscaRL boosts exploration and diversity, raising entropy (Figure 3b). We fix decay speed $\alpha=125$ and set the midpoint $t_0$ where entropy peaks near 1.2 to safely maximize exploration.
> * **Sensitivity**: Performance remains stable across a relatively wide range of hyperparameter choices. As shown in Figures 4c and 4d, even suboptimal settings generally score above 55.0, consistently outperforming the 52.0 Rubric-based RL baseline.

---

> > ### Author Rebuttal · Reviewer_cuDF · 2026-04-06
> >
> > Thanks for the detailed rebuttal. while these clarifications strengthen the work, certain concerns remain about using the same rubrics  across rollout generation and evaluation criteria for answer leakage problem and the limited testing to other tasks such as deep research tasks, long-context reasoning tasks and general reasoning tasks such as code and agent planning, tool use abilities.

---

> > > ### Author Response · Authors · 2026-04-08
> > >
> > > Thank you for the follow-up. We would like to clarify the remaining concerns as follows:
> > >
> > > - **On the "answer leakage" concern.** This is not the setting used in our evaluation. In our method, rubric-based scaffolding is used during rollout generation in training, while evaluation is conducted on held-out benchmarks **without** providing rubric scaffolding to the policy model. In addition, rubrics are **instance-specific**, rather than shared as a fixed answer template, and the rubrics used in training and evaluation are **not the same**. Therefore, this concern does **not** apply to our evaluation setup.
> > >
> > > - **On the "limited testing" concern.** We respectfully believe that our empirical study should not be characterized as limited. As noted in your strengths, "The empirical evaluation is extensive, covering multiple model families, varying parameter scales, and diverse task domains (medical, writing, instruction following, STEM)." In this light, we view the current experimental scope as already broad for the goals of the paper. Additional evaluations on deep research tasks, long-context reasoning, code, agent planning, and tool use would be valuable, but we see them as beyond the intended scope of the present work and more appropriate directions for future study.

---

### Official Review · Reviewer_6kSF · 2026-03-14

**Soundness:** 3
**Presentation:** 3
**Significance:** 2
**Originality:** 2
**Overall Recommendation:** 4
**Confidence:** 3

**Summary:**

This paper addresses the exploration bottleneck in reinforcement learning for LLM reasoning, particularly for complex open-ended tasks. The authors propose RuscaRL, which uses rubrics as explicit scaffolding injected into instructions during rollout generation, with the amount of scaffolding decayed over training and as verifiable reward signals via LLM-as-a-Judge. Experiments demonstrate the effectiveness of RuscaRL in improving reasoning performance on various benchmarks.

**Compliance With Llm Reviewing Policy:**

Affirmed.

**Final Justification:**

After reading the other reviewers' comments and the authors' responses, I decided to keep my original score.

**Key Questions For Authors:**

1. How is the rubric criteria generated? Is it generated by a human or a model? If it is generated by a model, how sensitive is the method to rubric quality?
2. What is the grader in the experiments? How does the choice of grader affect performance?

**Limitations:**

yes

**Strengths And Weaknesses:**

**Strengths**
- The paper is well-written and clearly motivated.
- The evaluation is rigorous. Experiments span multiple domains, multiple model families and scales, and include ablations on intra-group rubric ratio and inter-step decay.

**Weaknesses**
- Important implementation details regarding rubric construction and grader are insufficiently described in the paper. In particular, the process by which rubric criteria are generated and the degree to which rubric quality influences downstream performance are not adequately discussed.

---

> ### Author Rebuttal · Authors · 2026-03-31
>
> We sincerely appreciate the reviewer for highlighting the strengths of our work, including the clear presentation and motivation, the rigorous empirical evaluation, and the broad experimental coverage across multiple domains, model families, and scales.
> Below we address the main points raised in the review.
>
> **[W1,Q1]: Implementation details regarding rubric construction, grader, and sensitivity to rubric quality**
>
> Thanks for pointing out the need for these important implementation details.
> We have clarified these aspects in the manuscript:
> * **Rubric Construction:** As detailed in Appendix E.3, the rubrics are generated by GPT-4.1. For datasets with existing ideal answers, we use the question-answer pairs directly; for others, we first generate example solutions using GPT-4.1 and base the rubrics on those. The use of ideal answers or example solutions is designed to anchor good responses [1,2].
> * **Grader Details:** During training, we use Qwen3-32B (non-thinking) as the grader, which is noted in the Training Configurations (Appendix C.1). For evaluation (Appendix C.2), we use the officially recommended GPT-4.1 for HealthBench and Claude-Sonnet-4 for WritingBench and Creative Writing. For the Best-of-N evaluation, scoring 2048 samples per prompt with closed-source APIs is cost-prohibitive, so we utilized Qwen3-32B as a cost-effective alternative grader.
>
> **Sensitivity to Rubric Quality:** We evaluated the impact of rubric quality in Appendix D.8 by introducing various types of noise. As shown in Table R1 (Table 11 in the manuscript), RuscaRL maintains robustness against mild perturbations, though severe corruptions naturally degrade performance.
>
> Table R1. Robustness to rubric noise on medical benchmarks.
> |Rubric variant|HealthBench|LLMEval-Med|MedQA|MedMCQA|
> |:--|:--|:--|:--|:--|
> |**Initial Model**|23.4|48.0|61.8|56.3|
> |**+Rubric-based RL**|||||
> |├─ Original|52.0|56.5|62.3|56.3|
> |├─ Inverse|8.4|42.6|61.8|56.0|
> |├─ Negated|4.3|38.2|61.2|55.8|
> |├─ Ambiguous|43.7|55.6|62.1|56.1|
> |├─ Contradictory|46.8|56.2|62.0|56.0|
> |└─ 50% removed|43.2|54.8|61.9|56.2|
> |**+RuscaRL (Ours)**|||||
> |├─ Original|56.4|65.3|63.5|56.5|
> |├─ Inverse|11.9|45.7|62.1|56.2|
> |├─ Negated|8.2|42.8|61.5|56.0|
> |├─ Ambiguous|49.8|62.4|63.1|56.3|
> |├─ Contradictory|49.3|60.5|63.2|56.4|
> |└─ 50% removed|48.1|58.6|63.0|56.2|
>
> **[Q2]: The choice of grader and its effect on performance**
>
> Thanks for the valuable question regarding the impact of the grader choice. We have supplemented our paper with additional experiments on different graders and their alignment with human evaluations.
>
> Table R2 demonstrates how different grader models affect downstream medical performance when training Qwen2.5-7B-Instruct.
> Table R3 shows the agreement between LLM evaluations and human expert judgments across 940 expert-annotated criteria.
> As expected, stronger grader models generally exhibit higher agreement with human evaluators and lead to better downstream task performance.
> However, there are diminishing marginal returns; continuing to scale up the grader size (e.g., from Qwen3-32B to Qwen3-235B) yields limited additional benefits. Therefore, our choice of Qwen3-32B (non-thinking) offers an optimal balance between performance, human alignment, and computational cost.
>
> Table R2. Impact of different grader models on medical performance.
> |Grader|HealthBench|LLMEval-Med|
> |:--|:--|:--|
> |Qwen2.5-7B-Instruct|45.3|56.6|
> |Qwen3-30B-A3B-Instruct|48.9|60.9|
> |Qwen3-32B (non-thinking)|56.4|65.3|
> |gpt-oss-120B (auto)|56.6|65.0|
> |Qwen3-235B-A22B-Instruct|56.6|65.8|
>
> Table R3. Agreement between Human and LLM evaluations. (Cohen’s Kappa for inter-rater reliability. F1 Score treats human scores as ground truth).
> |Grader|Cohen's Kappa|F1 Score|
> |:--|:--|:--|
> |Qwen2.5-7B-Instruct|0.58|0.81|
> |Qwen3-30B-A3B-Instruct|0.74|0.90|
> |Qwen3-32B (non-thinking)|0.76|0.90|
> |gpt-oss-120B (auto)|0.74|0.88|
> |Qwen3-235B-A22B-Instruct|0.80|0.91|
>
> All these additional implementation details and the extended analyses on grader selection will be fully incorporated into the revised manuscript.
>
> ---
>
> **References**
>
> [1] Rubrics as Rewards: Reinforcement Learning Beyond Verifiable Domains. arXiv 2025.
>
> [2] RubricHub: A Comprehensive and Highly Discriminative Rubric Dataset via Automated Coarse-to-Fine Generation. arXiv 2026.

---

> > ### Author Rebuttal · Reviewer_6kSF · 2026-04-02
> >
> > Thank you for the detailed response and additional information about the rubric and grader. Since my original score is already positive, I will keep it as is.

---

> > > ### Author Response · Authors · 2026-04-06
> > >
> > > Thank you for reviewing our rebuttal and for your positive score. Your initial questions prompted us to provide a much clearer explanation of the rubric and grader mechanisms, which we believe significantly strengthens the manuscript. We greatly appreciate your constructive feedback and will include all these new details in the camera-ready version.

---

### Official Review · Reviewer_qZYi · 2026-03-24

**Soundness:** 3
**Presentation:** 3
**Significance:** 3
**Originality:** 3
**Overall Recommendation:** 3
**Confidence:** 4

**Summary:**

The paper proposes RuscaRL - a framework to break the exploration bottleneck for general LLM reasoning. RuscaRL integrates rubric-based instruction into the grading scheme by 1) incorporating itemized rubrics into the prompt instruction during generation, and 2) using LLM judge with the rubrics to provide a composite reward across different items inside a rubric during RL training. Experiments on multiple backbones (Qwen-3, -2.5 and Llama-3) are trained on HealthBench and evaluated on multiple Medical QA, instruction following and writing benchmarks against rubric-based RL, MeRF, RL-Plus, EA RL and SFT.

**Compliance With Llm Reviewing Policy:**

Affirmed.

**Key Questions For Authors:**

Why was comparison with baselines done on Qwen2.5 models only? The gains from other baselines already seem quite small, and I am curious to see the difference in performance when a much stronger base model is used (e.g., Qwen3)

**Limitations:**

Yes

**Strengths And Weaknesses:**

Strengths:
- The paper is well motivated as the exploration bottleneck problem is important and timely.
- The idea has promise and experiments are done thoroughly.

Weaknesses:
- I have doubts on whether the MedicalQA + instruction following dataset is the correct setting to suggest RuscaRL actually push beyond existing exploration bottleneck, as we see in Table 1 and 2 that the base model is already relatively good at the task.
- Missing a straightforward baseline would be simple GRPO on the same dataset

---

> ### Author Rebuttal · Authors · 2026-03-31
>
> We sincerely appreciate the reviewer for highlighting the strengths of our work, including the well-motivated focus on the important and timely exploration bottleneck problem, the promise of the proposed idea, and the thorough experimental validation.
> Below we address the main points raised in the review.
>
> **[W1]: Initial Model Performance and the Exploration Bottleneck**
>
> Thank you for this important comment.
> As stated in Section 5.2, "MedQA and MedMCQA are closed-ended multiple-choice benchmarks. Our improvements on these closed-ended benchmarks are marginal and are included only to demonstrate cross-task generalization." For instruction following datasets, we acknowledge that stronger models like Qwen3-30B-A3B-Instruct show modest gains since their capabilities are already relatively saturated, while we still observe substantial improvements on the Qwen2.5 and Llama-3 series.
>
> Accordingly, our exploration-bottleneck claim is mainly supported by the open-ended benchmarks (especially HealthBench and LLMEval-Med), together with the analyses in Section 5.3 and Appendix D.5:
> * **Capability Ceiling (Best-of-N):** The Best-of-N evaluation demonstrates that RuscaRL fundamentally raises the model's capability ceiling, rather than just improving sampling efficiency. We also report training dynamics including policy entropy and validation score.
> * **Novelty and Diversity:** The results further show that RuscaRL actively increases response diversity, and discovers responses that the original model finds nearly impossible to generate.
>
> **[W2]: Clarification on the Straightforward GRPO Baseline**
>
> We apologize for the confusion.
> We would like to clarify that our "Rubric-based RL" baseline is essentially the simple GRPO algorithm. As mentioned in Section 5.1 (Baselines), it is "implemented with the GRPO algorithm using rubric scores as rewards." Because our tasks are open-ended and lack verifiable ground-truth answers, we must use rubrics as the reward signal for any RL setup. We will explicitly state that "Rubric-based RL is simple GRPO" in the revised manuscript to prevent any future misunderstanding.
>
> **[Q1]: Baseline Comparisons on Stronger Base Models**
>
> Thanks for the suggestion.
> Due to compute and space constraints, our main baseline comparisons were focused on the Qwen2.5 series. To address your question, we have supplemented the experiments with stronger base models (Qwen3-4B-Instruct and Qwen3-30B-A3B-Instruct), comparing RuscaRL against the Rubric-based RL (GRPO) baseline. As shown in the table below, RuscaRL achieves substantial improvements over the baseline even on these stronger models, effectively demonstrating the power of our rubric scaffolding mechanism.
>
> |Model|Method|HealthBench|LLMEval-Med|
> |:--|:--|:--|:--|
> |Qwen3-4B-Instruct|Initial|40.2|66.7|
> ||+ Rubric-based RL (GRPO)|49.6|68.4|
> ||+ RuscaRL (Ours)|56.5|72.3|
> |Qwen3-30B-A3B-Instruct|Initial|46.9|71.5|
> ||+ Rubric-based RL (GRPO)|53.3|72.0|
> ||+ RuscaRL (Ours)|61.1|73.2|

---

> > ### Author Rebuttal · Reviewer_qZYi · 2026-04-04
> >
> > Thank you for the new experiments and clarifications. My concerns are mostly addressed, and i will update my score accordingly.

---

> > > ### Author Response · Authors · 2026-04-06
> > >
> > > Thank you for your time, constructive feedback, and for updating your score. We are very glad that our new experiments and clarifications have resolved your concerns. We will ensure that all the new results and discussions are fully incorporated into the final version of the paper.
> > >
> > > We noticed that the score hasn't been updated yet, so we were wondering if you might have forgotten, and we would really appreciate it if you could update it when you have some free time.

---

### Decision · Program_Chairs · 2026-04-30

**Decision:**

Accept (regular)

**Comment:**

**Summary:** This paper proposes RuscaRL, a reinforcement learning framework that uses checklist-style rubrics in a dual role: as explicit scaffolding injected into rollout prompts to guide diverse exploration, and as verifiable reward signals via LLM-as-a-Judge binary evaluation per criterion. The scaffolding is decayed over training via a sigmoid schedule, encouraging the model to internalize reasoning patterns rather than depend on external guidance. Experiments across medical, writing, and instruction-following benchmarks with Qwen2.5, Qwen3, and Llama3 backbones show consistent improvements, with a Qwen3-30B model matching OpenAI o3 on HealthBench-500.

**Summary of reviews:** Four reviewers scored this paper 3 to 4. All acknowledged the problem is well-motivated, the experiments are thorough, and the rubric-based design is clean and practical. Two reviewers marked their concerns as fully resolved after the rebuttal, one found the rebuttal helpful but retained a residual theoretical concern, and one maintained concerns about evaluation scope. The main residual issues are that the interaction between scaffolding heterogeneity and group-relative advantage normalization lacks full theoretical justification, and that gains outside the primary medical domain are more modest, making the "general reasoning" framing somewhat overstated.

**Assessment:** The paper makes a clear practical contribution to an important open problem: how to do RL on open-ended tasks where no verifiable ground truth exists. The rubric-as-scaffolding idea is not simply a reward engineering trick. The decay mechanism and intra-group differentiation of scaffolding strength represent a thoughtful design that balances exploration guidance against overfitting to explicit hints. The robustness analysis under five types of rubric corruption is unusually thorough for this area.

The rebuttal was effective. The RubricHub experiment (rubrics from heterogeneous model sources) addresses the concern about overfitting to GPT-4.1's rubric style. The clarification that "Rubric-based RL" is GRPO removes a baseline confusion. The quantitative scaffolding-advantage breakdown gives useful transparency.

The main limitation is that the strongest gains concentrate on the HealthBench-trained domain, with more modest improvements on writing and instruction-following benchmarks. The paper would benefit from training on a more diverse rubric corpus to better support the "general" claim. The theoretical gap identified by one reviewer concerning what objective RuscaRL actually optimizes when the best-performing ratio is not unbiased importance sampling deserves attention in a revision but does not undermine the empirical results.